# How to Talk to Your Classifier: Conditional Text Generation with Radar–Visual Latent Space

**DOI:** 10.3390/s25144467

**Published:** 2025-07-17

**Authors:** Julius Ott, Huawei Sun, Lorenzo Servadei, Robert Wille

**Affiliations:** 1Infineon Technologies AG, 85579 Neubiberg, Germany; huawei.sun@infineon.com; 2TUM School of Computation, Information and Technology, Technical University Munich, 80333 Munich, Germany; lorenzo.servadei@tum.de (L.S.); robert.wille@tum.de (R.W.)

**Keywords:** language–vision learning, classification, explainable neural networks, radar

## Abstract

Many radar applications rely primarily on visual classification for their evaluations. However, new research is integrating textual descriptions alongside visual input and showing that such multimodal fusion improves contextual understanding. A critical issue in this area is the effective alignment of coded text with corresponding images. To this end, our paper presents an adversarial training framework that generates descriptive text from the latent space of a visual radar classifier. Our quantitative evaluations show that this dual-task approach maintains a robust classification accuracy of 98.3% despite the inclusion of Gaussian-distributed latent spaces. Beyond these numerical validations, we conduct a qualitative study of the text output in relation to the classifier’s predictions. This analysis highlights the correlation between the generated descriptions and the assigned categories and provides insight into the classifier’s visual interpretation processes, particularly in the context of normally uninterpretable radar data.

## 1. Introduction

Sensors and devices tailored to the Internet of Things (IoT) are increasingly integrating into everyday lives. Among the countless applications, real-time estimation of the number of people is a particularly challenging and common problem. When used in private environments, such a feature can significantly improve intelligent energy and climate management by monitoring the occupancy of a specific space. During a pandemic, these technologies play a critical role in managing crowds and raising alarms to prevent the spread of infectious diseases. Different sensors, including Wi-Fi/Bluetooth [1], thermal [2], lidar [3], cameras [4], and radar, offer different approaches to accomplishing this task, each with its own limitations. For example, thermal sensors are susceptible to interference from sunlight, while Wi-Fi sensors, although designed primarily for communication, require extensive data processing to obtain physical measurements. Lidar sensors provide a high level of detail and are prone to problems in changing lighting conditions. Cameras that arguably perform better in indoor surveillance pose significant privacy concerns, preventing their use in smart home applications. Therefore, our focus is on radars, particularly frequency-modulated continuous wave (FMCW) radars, which are known for their resilience to weather and illumination, as well as their ability to maintain the secrecy of private data. Additionally, their economical processing capabilities enable the detection of a person’s distance, angle, and speed, although the resolution of the radar limits the number of people that can be detected. In this study, we utilize a cost-optimized radar with just three receiving antennas, which can be used for elevator surveillance or door bells. Such systems have to run all the time; thus, limiting power consumption is inevitable. Related works on radar focus on high-resolution tasks with more antennas, like pose estimation [5,6,7] or automotive long-range detection with higher frequencies [8].

The data collected by radars are often converted into an image format, a format with respect to which neural networks have demonstrated outstanding capabilities [9]. Recent breakthroughs in both processing chains and neural network design have demonstrated the effectiveness of FMCW radars in people counting, with an accuracy of up to six people. To ensure the viability of these systems in real-world environments, it is critical to consider a variety of scenarios and environmental conditions during their development. Factors such as individual movement patterns and radar mounting height are crucial. For example, the Range-Doppler image differs depending on whether people are moving or standing. If there are multiple people in the radar range, the system must detect whether they are stationary or moving. Studies have taken these variables into account and trained and tested algorithms in controlled environments [10], which were then adjusted to work properly in different environments [11].

However, for systems to be truly effective, they must also be interpretable. Conventional methods strive to elucidate decisions made by neural networks at the input level through attention maps [12]. These techniques are too computationally intense, and radar images are inherently complex, making attention maps indecipherable by an untrained user.

To close this gap, a system that can provide understandable interpretations to any user in real time must be developed. Using a description of the neural network’s interpretation through text can result in a generally understandable explanation, similarly to creating captions. The challenge lies in combining radar data with text descriptions.

A typical approach involves training an image-encoding network with a text generation network simultaneously, where popular image-to-text models use encoder and decoder Transformer structures to learn shared representations [13,14]. Despite advances in text generation and image captioning, Transformers have not surpassed Convolutional Neural Networks (CNNs) in processing sensor data due to the latter’s lower data requirements. In [15], radar and image embeddings are aligned for text generation under difficult conditions for cameras, namely darkness. Like many other radar studies, they use radars that consume too much power for consumer radar applications. This work uses two radars with 12 antennas each. Similarly, our research focuses on converting the latent representation of a CNN-based classifier into text for the improved interpretation of the classifier’s decision. However, simply attaching an LSTM for text creation does not meet the explainability requirements. We aim for a system where small changes in the latent space correspond to small changes in the resulting text, which should also correlate with the exact or expected number of people.

The solution to this problem lies in the domain of the Variational Autoencoder (VAE), which maps inputs into a latent Gaussian space for reconstructing initial data. By utilizing this architecture, we can harness the latent classifier space for projections into the Gaussian domain and generate text. Such an adaptation involves substituting the visual encoder within the VAE, ensuring that no gradient from text reconstruction permeates the classifier. However, as the literature indicates [16,17], the VAE has limitations, namely, overlooking the latent representation and lacking encoded structure—where a structured latent space should align with the nuances of the reconstructed text. The incorporation of a denoising target in the Denoising Adversarial Autoencoder (DAAE) introduces structure by reconstructing unaltered sentences from their perturbed counterparts, while adversarial training schemes emphasize the Gaussian latent representation for reconstruction purposes. The application of such adversarial training in the context of classification, particularly for radar sensors, has yet to be explored. Moreover, the potential limitations on the classifier’s performance posed by the latent space and scalability considerations are also subjects of our ablation study. The generated descriptions are evaluated using the ROUGE-L score [18], which measures the overlapping sequence length between reference and generated descriptions.

To summarize, the goal of this work is to show how text can be aligned a classifier without losing accuracy, and the detailed contributions of this work are as follows:We propose a training method, combining a visual classifier with a DAAE, where the DAAE is tasked with reconstructing text captions corresponding to radar images.We show the capacity of the presented method to generate radar image descriptions via the classifier’s latent representation, thereby enhancing interpretability and ensuring alignment with the classified outcome.We confirm, through an ablation study, that our method of text generation does not interfere with classification efficacy (98.3%), even when adjustments are made to increase the force on the Gaussian constraint.

Our experiment with an industrial radar dataset produced promising results, indicating that it is possible to generate reliable text that reflects visual interpretations while preserving the integrity of classification performance. This framework significantly improves the usability of neural networks in real-world scenarios, especially when combined with sensor data that would otherwise be cryptic to human interpretation.

This manuscript addresses the existing discourse on the explainability of neural networks, the synergy of image–text learning, and latent space modeling, and then, it explains the DAAE approach and the mechanisms for generating text descriptions from visual latent representations. In Section 2, seminal work on text generation and latent space modeling is reviewed. Afterward, the building blocks for this work are presented in Section 3, such as dataset generation in Section 3.1, the DAAE in Section 3.2, and the model’s design for text reconstruction in Section 3.3. The method is evaluated on an industrial radar dataset with respect to classification accuracy and text quality in Section 4 and then summarized in Section 5.

## 2. Related Work

In this section, we address the basic research underlying the three core aspects of our proposed approach: the explainability of neural networks, the integration of visual and semantic information, and the manipulation of latent space to control text generation.

### 2.1. Explainability of Neural Networks

The rise of complex neural network (NN) models in areas such as computer vision (CV) and natural language processing (NLP) has created an urgent need for explainable artificial intelligence (XAI). XAI is critical for detecting biases within datasets and ensuring that algorithms are fair and transparent. Various methods have been developed to decipher how neural networks work. One notable approach is to leverage a network’s internal processing to shed light on how particular inputs produce their respective outputs. One technique, Local Interpretable Model-Agnostic Explanations (LIME) [19] simplifies this by approximating the complex model with a more interpretable surrogate model that can analyze the meaning of each input feature. Similarly, Shapley Additive explanations (SHAPs) [20] calculates the contribution of each input feature to the output of the network. Other methods such as Class Activation Mapping (CAM) [21] and Gradient-Weighted Class Activation Mapping (Grad-CAM) [22] use neural sensitivity and dimensionality to generate saliency maps that express feature relevance in high-dimensional spaces. However, the complexity of these functional spaces makes it difficult to interpret networks in real time. Meanwhile, transformer networks, with their innate attention mechanisms, provide insight into their decision-making processes [23].

Besides finding the relevant input features, explainability can be obtained from the representations of the network. Techniques like Principal Component Analysis (PCA) [24] or Independent Component Analysis (ICA) [25] extract low-dimensional representations from the neural network’s embedding space to show which inputs are entangled in the latent space. The desired class separation can also be learned. In InfoGAN [26], an adversarial training scheme aims to reduce the entanglement of representations. This effect can also be achieved with specific loss functions [27].

However, these explanatory techniques are most useful when dealing with human-interpretable inputs such as images or text. These methods fail when it comes to data that are not immediately understandable to humans—such as frequency signatures from WiFi, radar, or ultrasound sensors. Therefore, to explain such data, we try to use multimodal approaches, such as visual–semantic learning.

### 2.2. Visual–Semantic Learning

The goal of visual–semantic learning is to create a harmonious relationship between images and their textual descriptions. Research in this area has made significant progress, and this is exemplified by studies such as [28], which label visual entities and map them into a shared latent space with text using dual networks called projectors. CLIP [14] further advances this concept by combining images with text in a common embedding space trained on large-scale image–text pairs sourced from the Internet. Recently, this idea has also been extended to radar data, learning alignments between radar point cloud and text descriptions [29]. This methodology has been applied to a variety of tasks, including image captioning [13,30], image–text retrieval [31], and even generative image creation tasks based on text input [32].

Image captioning models are the closest to our vision of a language-explained visual model, as they infer text descriptions based on visual signals. These approaches differ in the integration of image and text information—some use soft prompts [33], and others use a trainable text generation network [13]. However, the usefulness of these models depends on the accuracy of the image encoder. Inaccuracies in the visual interpretation make the captions nonsensical. Here, the role of latent space is crucial: it must maintain its semantic integrity regardless of the changes within it.

### 2.3. Latent Space Modeling

Text generation methods such as [34] can create modified text by manipulating the latent representation of the language model. This allows for advanced grammatical changes or linguistic style transfers, as shown in [35]. These methods learn specific mappings for latent space manipulation or use keyword searches to change the value vectors of specific transformer layers. When it comes to random text generation, ref. [36] reported an invertible function of the latent distribution to a Gaussian prior. In this way, perturbations in the prior space are mapped to the latent distribution and allow the flexible control of the generated text. Similarly, VAEs are used in [37] to estimate the latent space of a Bert-GPT2 [38,39] encoder–decoder model with a Gaussian prior steer. This highlights that Gaussian priors are promising for text manipulations in latent representations. However, the seminal works focus mostly on transformer-based text manipulation, and VAEs suffer from ignoring the latent prior [16].

Concurrent work proposed a DAAE that is trained in an unsupervised fashion by reconstructing perturbed sentences [40]. The DAAE is conditioned on a prior Gaussian distribution and assumes noisy inputs. Thus, the DAAE fulfills two important criteria to enable the interpretation of a visual classifier: (1) The classifier’s latent space can be modeled as a Gaussian distribution from where the decoder of the DAAE can generate text. (2) Given a description of the model, the DAAE learns to reconstruct the caption by focusing on the latent representation of the classifier through the adversarial training objective. (3) The DAAE interprets the visual latent space as a noisy version of the true latent representation, compensating for an imperfect classifier.

## 3. Approach

The approach is divided into three parts. First, we describe the radar dataset. Afterward, we introduce DAAE, and then, we demonstrate how it can be used to generate text from a visual classifier embedding. In this way, the text explains the visual interpretation of the radar input.

### 3.1. Radar Signal Processing

The proposed method uses Infineon’s BGT60TR13C FMCW radar chipset (Infineon Technologies AG, Neubiberg, Germany), as shown in Figure 1. This chipset operates at a base frequency of 60 GHz and has one transmit antenna (TX) and three receive antennas (RX). With an ultra-wide bandwidth of 5.5 GHz, a very high distance resolution of up to 3 cm, and a ramp-up speed of 400 MHz/μs, a higher Doppler speed can be achieved. In addition, the high signal-to-noise ratio (SNR) ensures the detection of people up to 15 m when in front of the sensor, while its high sensitivity enables the detection of sub-millimeter movements. Thanks to optimized performance modes during sensor operation, the lowest power consumption of less than 5 mW can be guaranteed. These interruptions serve to save energy and facilitate data preprocessing. The collected data are digitized to 12 bits by the analog-to-digital converter (ADC) and then forwarded from the evaluation board via USB to the PC for further examination. We can use the radar signal to calculate the range of individual targets. The distance resolution (ΔR) and maximum detectable range ($R_{max}$) can be calculated using the following formulas [41]:(1)$$\Delta R=\frac{c}{2B},$$(2)$$R_{max}=\frac{\Delta R}{2}N_{s},$$
where *c* denotes the speed of light, *B* denotes the bandwidth, and $N_{s}$ denotes the number of samples. In this setting, we use a bandwidth of 1 GHz, which results in a resolution of 15 cm. In combination with 128 samples per chirp, a maximum range of approximately 10 m is enabled, which is particularly suitable for indoor observations. In order to measure the velocity of the targets, we use the Doppler frequency along with the number of chirps. The velocity resolution (ΔV) and the maximum detectable velocity ($V_{max}$) are computed as follows:(3)$$V_{max}=\frac{c}{4f_{0}t_{c}},$$(4)$$\Delta V=\frac{2V_{max}}{N_{c}},$$
where $f_{0}$ denotes the center frequency, $t_{c}$ denotes the chirp time duration, $N_{c}$ denotes the number of chirps, and *c* denotes the speed of light [41].

We set the chirp time to 391 μs; thus, we can detect up to 3.2 m/s and resolve 0.1 m/s. With an average human walking speed of 1.42 m/s, we can detect all sorts of human motion. The configuration of the used radar and the relevant parameters are detailed in Table 1.

The received frame in each interval takes the form of a three-dimensional array for which its dimensions ($N_{c}$, $N_{s}$, and $N_{rx}$) are based on the number of chirps ($N_{c}$), the number of samples per chirp ($N_{s}$), and the number of receiving antennas ($N_{rx}$). The axis along the chirps represents slow time, while the axis along the samples represents fast time. To avoid leakage from the transmit/receive antenna, we subtract the average in the fast time. Signal processing occurs on two separate chains. In a setup designed to sense human presence, a person can display significant movements, such as walking or running (macro movements), and subtle movements, such as breathing, or slight body movements while standing still (micro-movements). First, we use the coherent integration of $N_{c}$ consecutive frames to create a virtual frame that increases the SNR of micro-motions (breathing) [42]. At the same time, macro-movements are captured in the latest image. The following processing steps include a moving target indicator (MTI) to remove reflections from completely static targets, followed by a 2D fast Fourier transformation (FFT). For the MTI, we subtract the average along the slow time axis, as applied in [42]. After these processing steps, we obtain two RDIs for macro- and micro-motion and stack them to obtain a radar input with two channels. The macro- and micro-frame processing steps for each antenna are shown in Figure 2.

The obtained radar data, as described in Section 3.1, capture scenes of up to five people performing three different movements: walking, sitting, and standing in six different office rooms. The data are randomly split by recording in two datasets: one for training and one for testing. The training data contain 300,000 frames, which comprise 8 h of recorded radar data at a frame rate of 10 Hz. The training classes are distributed as follows: 27,181 with zero people, 53,391 with one person, 44,701 with two people, 71,824 with three people, 61,186 with four people, and 41,717 with five people. The test dataset contains 90,000 frames. The test classes are distributed as follows: 10,296 samples with zero people, 15,268 with one person, 15,623 with two people, 19,763 with three people, 17,215 with four people, and 11,835 with five people. In each of the training recordings, 20% of the frames are used for validation, and the remaining 80% are used for training. Afterward, the presented preprocessing steps are applied to prevent any information leakage through the micro-frame buffer.

In addition, each recording has a description that describes the motion of individuals, e.g., “three people walking and two people sitting”. The considered movements are walking, sitting, and standing.

### 3.2. Denoising Adversarial Autoencoder

The DAAE of [40] extends the concept of an Adversarial Autoencoder (AAE) [17,43] by incorporating a local perturbation process *C* into the input data during training. These perturbations corrupt the original input data x∈X to $\tilde{x}\in X$. Let the joint probability of the data be $p\left(x,\tilde{x}\right)=p_{data}\left(x\right)p_{C}\left(\tilde{x}|x\right)$ and the marginal probability be $p\left(\tilde{x}\right)=\sum_{x}p\left(x,\tilde{x}\right)$. The model’s task is to reconstruct the original data from the perturbed version. The individual building blocks are an LSTM for text encoding $E_{text}$, an LSTM for text generation *G*, and a single-layer discriminator *D*. The encoder $E_{text}$ is later replaced by the visual encoder. This is achieved by the following training objectives:(5)$$\min_{E,G}\max_{D}L_{rec}\left(\theta_{E},\theta_{G}\right)-\lambda L_{adv}\left(\theta_{E},\theta_{D}\right),$$
with(6)$$\begin{array}{cc} L_{rec}\left(\theta_{E},\theta_{G}\right) & =E_{p(x,\tilde{x})}-\log p_{G}\left(x|E_{text}\left(\tilde{x}\right)\right) \end{array}$$(7)$$\begin{array}{ccc} L_{adv}\left(\theta_{E},\theta_{D}\right) & = & E_{p(z)}\left[-\log D\left(z\right)\right]+\\ & & E_{p(\tilde{x})}\left[-\log\left(1-D\left(E_{text}\left(\tilde{x}\right)\right)\right)\right], \end{array}$$
where $L_{rec}$ is the reconstruction loss for the estimated $\tilde{x}$ with respect to the input *x*, and $L_{adv}$ is the adversarial loss evaluating the latent distribution of the encoded $\tilde{x}$. This setup follows a typical Generative Adversarial Network (GAN) training method, where the discriminator and generator interplay in an actor–critic fashion. However, the generator and discriminator act on the latent space to enforce the Gaussian distribution, as in the VAE. Thus, reconstruction loss is needed for high text quality. This approach encourages the network to map sequences to appropriate latent spaces without the need for additional training objectives or reparameterization-style tricks like VAE. Furthermore, DAAE overcomes the VAE by neglecting the latent information combined with noisy inputs, as shown in [16]. In addition, encoders and decoders are LSTM blocks and not transformers. Better performance with LSTM blocks was shown by [40]. Although LSTMs are known to perform poorly for very long text sequences, this issue is not relevant in this particular use case. Below, we can replace the encoder with a visual classifier that presents a noisy view of the original input.

### 3.3. Decoding Text from Image Classifier

In this section, we show how text can be decoded based on the latent representation of the visual classifier. The focus is on training a classifier to count up to five people. At the same time, we want to generate text from the classifier embedding. In this way, the text describes the classifier’s interpretation of the radar data. The text generator will be trained independently of the classifier. The overall architecture of the visual classifier and text decoder is shown in Figure 3.

The visual classifier consists of a CNN-based encoder where each block processes the macro- and micro-RDI individually and then uses a cross-convolution along the combined features. The radar image encoder is followed by two linear layers that encode the mean and standard deviation of the Gaussian latent representation. The visual encoding is defined as $E_{v}$.

With this design, the visual classifier has the same latent structure as the DAAE. Further, the joint latent structure enables the reconstruction of radar scene descriptions from the visual latent space using the decoder of the pretrained DAAE. Finally, a classification layer predicts the amount of people in front of the radar sensor.

Given the stochastic nature of classifier training and incomplete classification performance, the visual latent representation is a noisy version of the original input image representation. The training goal is to match the semantic description $\tilde{x}$ with the latent interpretation *z* of the radar’s input. This is achieved with the following loss functions:(8)$$\begin{array}{cc} \min_{E,G}\max_{D} & L_{cls}\left(\theta_{E_{v}}\right)+L_{rec}\left(\theta_{E_{v}},\theta_{G}\right)-\lambda L_{adv}\left(\theta_{E_{v}},\theta_{D}\right), \end{array}$$
with(9)$$\begin{array}{cc} L_{cls}\left(\theta_{E_{v}}\right) & =E_{p(x)}[-\log p_{cls}\left(c|E_{v}\left(x\right))\right] \end{array}$$(10)$$\begin{array}{cc} L_{rec}\left(\theta_{G}\right) & =E_{p(x,\tilde{x})}\left[-\log p_{G}\left(\tilde{x}|\bar{E_{v}}\left(x\right)\right)\right] \end{array}$$(11)$$\begin{array}{ccc} L_{adv}\left(\theta_{E_{v}},\theta_{D}\right) & = & E_{p(z)}\left[-\log D\left(z\right)\right]+\\ & & E_{p(\tilde{x})}\left[-\log\left(1-D\left(E_{v}\left(x\right)\right)\right)\right], \end{array}$$
where $\overset{-}{E_{v}}$ denotes the stop gradient operator. The proposed approach trains the vision encoder with the classification loss ($L_{cls}$) and enforces the Gaussian latent structure with adversarial training ($L_{adv}$) using the pretrained discriminator from the DAAE. The reconstruction loss can be interpreted as the alignment between radar and text features. However, reconstruction loss is used to fine-tune the text decoder without training the classifier. Since the focus is on the interpretation of the visual classifier, it should not be influenced by the text generator. Only the discriminator’s loss is necessary to enforce a common latent space structure. The Gaussian constraint must be carefully tuned, as we expect it to reduce classification performances.

## 4. Experiments

In this section, we first review the implementation settings. Afterward, we quantitatively benchmark the joint vision–language training with generated captions on an industrial radar dataset. Finally, we demonstrate the descriptive capabilities of the generated text.

### 4.1. Implementation Settings

In the implementation, we used PyTorch v2.0.^™^-GPU v2.4.0 with CUDA^®^ Toolkit v11.8.0 and cuDNN v8.5.0. As a processing unit, we used a single Nvidia^®^ Tesla^®^ P40 GPU, Intel^®^ Core i7-8700K CPU, and DIMM 16 GB DDR4-3000 module of RAM.

The DAAE is pre-trained on a review dataset from websites like Yelp and TripAdvisor [44]. The configuration and dataset are the same as those described in [40]. For the visual classifier, we used three cross-convolution blocks described in [45] as the backbone, with two connected dense layers for the mean and variance of the Gaussian latent representation and a final dense layer for the classification.

For joint training, we keep the vocabulary of the pre-trained DAAE comprising 10,000 words, which could result in words in captions that are not part of the vocabulary, but they are matched to related words. The joint architecture is trained with the Stochastic Gradient Descent (SGD) optimizer and an initial learning value of 0.05. The learning rate decays after 5, 15, and 25 epochs with a decay rate of 0.1. In addition, it is noteworthy that the networks are trained with 64 samples per batch on a single GPU for 50 epochs. The classification results are averaged over five different seeds.

The implementation is published on GitHub, with publicly available image datasets and automatically generated captions (a reproducible implementation of our method on public image classification datasets with automatically generated captions. Github link: https://github.com/juliusott/talk-to-your-classifier, (accessed on 12 June 2025)) provided for reproducibility.

### 4.2. Results

This section presents the experimental results of the proposed joint vision–language model. First, the classification results of the proposed method on the industrial radar dataset (see Figure 4) are shown. In addition, we demonstrate the effects of the different scaling factors of adversarial training objects that enforce the embedding space’s constraints. Second, we demonstrate the decoded text from the classifier embeddings, with a focus on texts where the classifier prediction is wrong.

#### 4.2.1. Classification

In sensor applications, the emphasis is on performance. As mentioned in Equation (Equation 8), the Gaussian constraint is enforced by the adversarial loss function with scaling factor $\lambda_{adv}$. The ablation study compares scaling factors $\lambda_{adv}=1,10,20$ to measure the interplay between Gaussian constraint and text reconstruction. As a baseline, we use an end-to-end classifier of the same architecture without the Gaussian latent space. Note that the classifier is not influenced by the text reconstruction’s loss, and hence, we only ablate force on the Gaussian constraint. The results illustrate that the classifier performance is minimally affected by the Gaussian latent space constraint and also achieves the same accuracy as the baseline. However, giving more importance to the Gaussian structure will result in slower convergence.

A similar conclusion can be drawn from Table 2, which demonstrates high overall performance, while it drops when the focus on the Gaussian constraint is too high. At the same time, we calculate the ROUGE score, which assesses how often words appear in the reference and model output. In particular, the ROUGE-L score is considered since it measures the longest common sequence between generated texts and references.

Table 2 shows how $\lambda_{adv}$ leverages classification accuracy and text quality. The best classification accuracy is achieved by reducing the Gaussian constraint. However, the ROUGE-L score shows how classification performance influences the text’s quality, which is reasonable since ambiguous features lead to poor text descriptions. The ROUGE score is the highest for $\lambda_{adv}=1$, but it decreases when $\lambda_{adv}$ is decreased or increased. In addition, the ROUGE-L score decreases for $\lambda_{adv}=10$ and increases again for $\lambda_{adv}=20$. This phenomenon can be explained by the neglect of the classifier and the heavy focus on the discriminator’s loss. The high score for the highest $\lambda_{adv}$ is attributed to simple words like “people”, which appear in all text descriptions.

Finding the best $\lambda_{adv}$ depends on the application, but we found that $\lambda_{adv}=1$ achieves the highest classification performance and the best ROUGE-L score.

#### 4.2.2. Decoded Scene Descriptions

Besides the classification performance, the main focus of this work is to provide a method for generating text descriptions that increase the interpretability of the classifier.

High-dimensional classifier embeddings are projected via t-distributed stochastic neighbor embedding (TSNE) [46]. In Figure 5, we examine the first and second TSNE components of the predicted latent representation for three-person radar images, where $\lambda_{adv}=1$ during training. The visualization focuses on the different movements that the three targets perform, namely, “walking,” “sitting,” and “sitting and walking.” As expected, the sitting and walking people are well separated, and the combination of walking and sitting people overlaps with the others. However, we also observe that different movements are sometimes very close in the classifier’s embedding space. This finding underlines that certain movements can have overlapping frequency signatures. To evaluate the descriptions, we consider examples where the classifier predicted the wrong number of people and the correct number of people. Figure 5 shows four predicted targets in red and five predicted targets in blue, while the actual number of people is still three.

Table 3 shows the different types of text descriptions that were observed. We compare the ground truth description with the reconstructed description and the predicted label. The main observation is that the reconstructed descriptions are aligned with the classifier’s prediction, which allows the interpretation of the classifier’s reasoning. There are also cases where the text description and classifier do not match.

These scenarios can be summarized into two cases: First, the text description is correct with respect to the ground truth, but the classifier is wrong; second, both predict wrong labels. In the latter case, the text description does not mention the number of people but describes the movements in the scene. The few mismatches can be explained by the interpretation of the latent space and the separation between classification and text reconstruction loss functions.

Overall, all different text descriptions provide a meaningful visual interpretation. To further illustrate the capabilities of the presented approach, the experiments were extended using images in Appendix A.

## 5. Conclusions

In this article, we prove how a DAAE can be used in combination with a visual classifier to provide instance interpretability with a classification performance of 98.3%. To achieve this, we recorded a large industrial radar dataset that takes different movements of the targets into account. Radar data requires latent space interpretability because the processed radar data is not human-interpretable. The results presented show that the common training goal of text reconstruction and classification not only leads to the same classification performance but also enables the interpretability of arguments. In addition, the text matches the predicted class and provides meaningful descriptions of the radar input. Furthermore, this method can be generalized across multi-domain tasks that can include video information as well. Finally, further research will focus on unreliable state detection. 

## Figures and Tables

**Figure 1 sensors-25-04467-f001:**
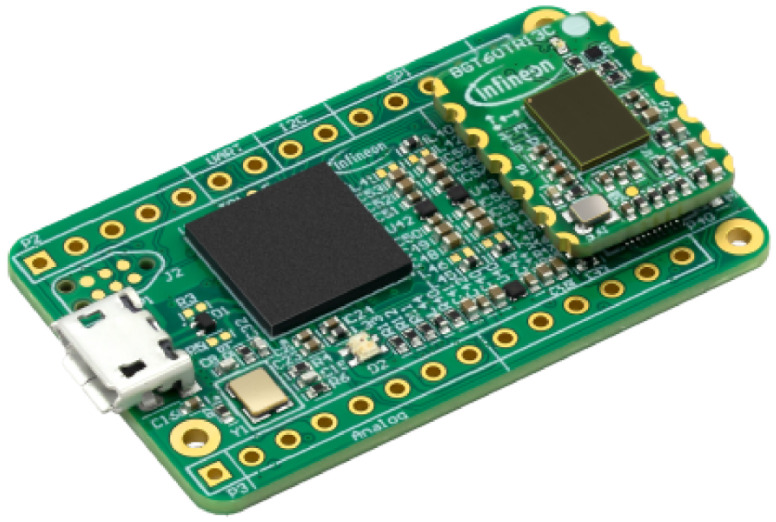
Infineon’s FMCW radar system BGT60TR13C with a size of 6.5 × 5.0 × 0.9 mm^3^. The system is based on a microcontroller that filters the radar signal, digitizes it, and transmits it to a computer via micro-USB.

**Figure 2 sensors-25-04467-f002:**
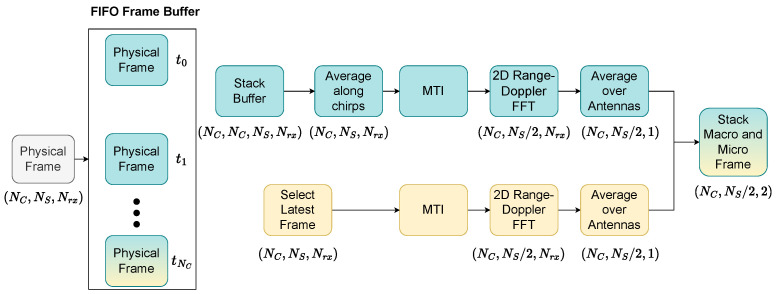
Radar processing steps from recorded radar frames to final macro–micro-RDIs. The steps in blue refer to the micro-movements and the steps in yellow refer to the macro-movements. The changing data form is mentioned after each processing step.

**Figure 3 sensors-25-04467-f003:**
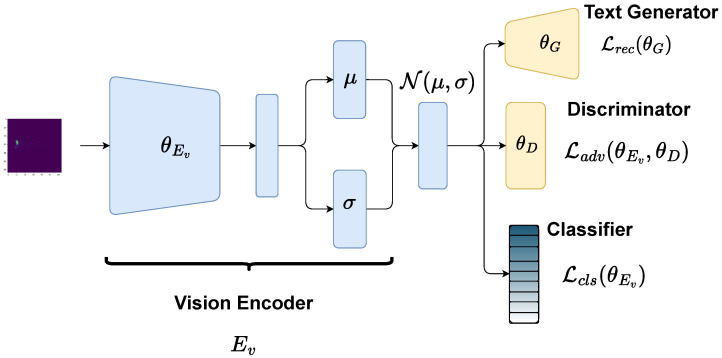
Joint architecture for text generation from a visual latent space. The vision encoder generates the mean and standard deviation of the Gaussian latent space. After reparametrization, the embedding is used for classification and text generation, while the discriminator makes sure the latent space is from a standard normal distribution. The yellow building blocks are pretrained parts from the DAAE.

**Figure 4 sensors-25-04467-f004:**
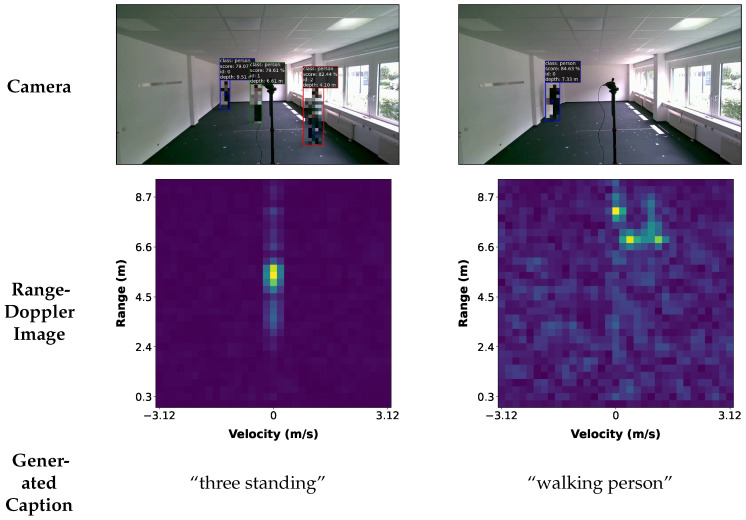
Illustration of the recording setup. The recorded scenes capture different numbers of people in an office environment. The sensor is mounted in front of the camera. Below the camera images, we show the corresponding Range-Doppler image and the generated caption.

**Figure 5 sensors-25-04467-f005:**
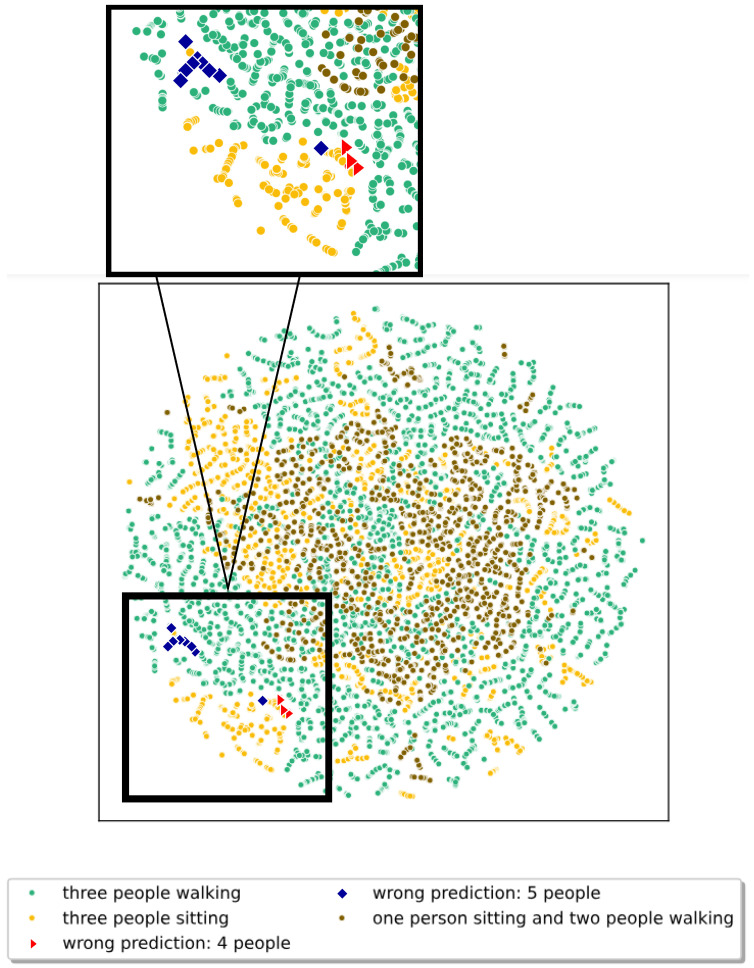
Visualization of first two TSNE components of the validation dataset where three people are present in a scene.

**Table 1 sensors-25-04467-t001:** Detailed radar configuration used throughout this work.

Symbol	Parameters	Value
$f_{0}$	Center frequency	60 GHz
*B*	Bandwidth (*B*)	(60.5–61.5) GHz
$N_{s}$	Number of samples per chirp ($N_{s})$	128
$N_{c}$	Number of chirps	64
$f_{s}$	Sampling frequency ADC	2 MHz
$t_{c}$	chirp time duration	390 μs
$t_{s}$	Frame repetition time	0.05 s
$N_{rx}$	Number of receiving antennas	3

**Table 2 sensors-25-04467-t002:** Test accuracy for different adversarial scaling factors and the respective ROUGE-L scores for text evaluation.

Model	Test Accuracy in %	ROUGE-L
Classifier (baseline)	98.31	-
Classifier + DAAE ($\lambda_{adv}=0.1$)	98.3	30.1
Classifier + DAAE ($\lambda_{adv}=1$)	98.3	30.8
Classifier + DAAE ($\lambda_{adv}=10$)	98.23	26.64
Classifier + DAAE ($\lambda_{adv}=20$)	98.02	30.57

**Table 3 sensors-25-04467-t003:** Comparison of ground truth text, reconstructed text, and the predicted label.

Ground Truth Text	Reconstructed	Classifier Predictions
“three people sitting”	“four walking”	4
“three people sitting”	“three sitting”	4
“three people sitting”	“four walking”	5
“three people walking”	“five walking”	5
“three people sitting”	“five sitting”	5
“three people sitting”	“person sitting and walking”	5

## Data Availability

The datasets presented in this article are not available because of company policies.

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
