# Peer review of "How to Talk to Your Classifier: Conditional Text Generation with Radar–Visual Latent Space"

_sensors, 2025, doi:10.3390/s25144467_

Round 1
Reviewer 1 Report
Comments and Suggestions for Authors
R e v i e w
Title: How to Talk to Your Classifier: Conditional Text Generation with Radar-Visual Latent Space
Authors: Julius Ott, Huawei Sun, Lorenzo Servadei, Robert Wille
Journal: Sensors (MDPI)
Aim. The paper aim is to present the framework that enables a radar-based visual classifier to generate textual descriptions by leveraging a shared radar-visual latent space.
Method. The offered method combining a Convolutional Neural Network (CNN)-based radar image encoder with a Denoising Adversarial Autoencoder (DAAE) trained to reconstruct text from perturbed input. This architecture enables the classifier’s latent space to serve as a semantic basis for conditional text generation — effectively “explaining” its decision in natural language.
Validation. Classification Accuracy: Up to 98.3% on an industrial radar dataset with over 300,000 training and 90,000 test frames. Text Generation Quality: Evaluated using ROUGE-L, with best scores (~30.8) achieved under minimal adversarial constraints (λ_adv = 1). Robustness: An ablation study shows that classification accuracy remains stable under different degrees of Gaussian constraint enforcement. Visualization: TSNE projections demonstrate meaningful separation in latent space based on the number of people and their movements.
The paper presents an original approach to improving the interpretability of neural networks applied to radar sensor data. The authors propose a multimodal learning architecture that combines:
- A CNN-based radar image classifier trained on FMCW radar data to detect and count up to five people based on movement patterns.
- A Denoising Adversarial Autoencoder (DAAE) pretrained on general-purpose review datasets to generate descriptive text from latent embeddings.
- A shared latent Gaussian space through which the classifier’s internal features are aligned with the text generation process.
The core idea is to “talk to classifier” by decoding its internal representations into human-readable captions such as “three people sitting” or “four walking”. This system aims to make otherwise opaque radar-based AI models more transparent and trustworthy—especially in contexts like smart homes, crowd monitoring, and privacy-sensitive environments.
My Comments for Editors and Authors:
I believe the topic addressed in this paper is relevant and timely. The research presented will likely be of interest to the readership of Sensors (MDPI).
The paper is recommended for publication Sensors (MDPI). However, I believe the manuscript should be revised, improved before accepting.
Here are my remarks.
remark 1:
It should be pointed the aim of this research in Introduction of the paper.
remark 2:
It should be described the structure of this research in Introduction of the paper.
In Sec. 1 .... In Sec. 2 .... In Sec. 3 .... etc.
remark 3:
The Introduction section of the text should use paragraph indentation.
remark 4:
The paper should include a Conclusion with an analysis of the obtained results and a description of plans for further research.
remark 5:
I believe that the phrase “visual radar classifier descriptive text” (Lines 6–7) is grammatically incorrect and needs clarification.
remark 6:
The limitations of LSTM for text generation are not discussed.
remark 7:
Loss functions are defined but not clearly explained in terms of practical intuition.
remark 8:
Only three λ_adv values are tested—lacks granularity in hyperparameter tuning. Why? It would be better to consider a larger quantity. In this case, the dependence on the parameter would become more apparent.
remark 9:
I believe that the phrase “barely influences” (Line 334) is vague; how much degradation is acceptable?
remark 10:
The paper claims this is the first attempt at radar-text alignment—no literature survey confirms that.
remark 11:
ROUGE is introduced without specifying which version (L, 1, 2, etc.).
remark 12:
The tables are formatted without regard to the journal’s requirements. (see template)
remark 13:
“We first, describe” (Line 178) contains a grammatical error—comma should be removed.
remark 14:
Grammar issue: “depends on whether all people are moving or some are standing” (Line 44) - consider rephrasing.
remark 15:
The quality of Figure 4. should be improved.
remark 16:
I believe that the Experiments section should be significantly expanded to demonstrate how the proposed approach performs in a more general setting. For example, when the number of people is not known in advance and varies over time. It would also be interesting to see the results obtained by the authors when the number of people is zero.

Author Response
Dear Reviewer,
thank you very much for the insightful suggestions to improve our paper. In the following, we will adress each remark individually and the respective changes are highlighted in red in the revised manuscript,
Comments 1: It should be pointed the aim of this research in Introduction of the paper
Response 1: Thank you for the suggestion, we added the goal of the work besides our contributions (line 90 and 91).
Comments 2: It should be described the structure of this research in Introduction of the paper. In Sec. 1 .... In Sec. 2 .... In Sec. 3 .... etc.
Response 2: We thank the Reviewer for his advise and clarified the structure of the paper alongside the numbers of each section in line 109-114.
Comments 3: The Introduction section of the text should use paragraph indentation.
Response 3: Thank you for the remark, we added the paragraph indentation in the introduction.
Comments 4: The paper should include a Conclusion with an analysis of the obtained results and a description of plans for further research.
Response 4: We sincerely apologize for this mistake and added the conclusion in line 387 on the last page.
Comment 5: I believe that t (Lines 6 7) is grammatically incorrect and needs clarification.
Response 5: Thank you for the remark. We rephrased the sentence in line 4-6 for clarity.
Comment 6: The limitations of LSTM for text generation are not discussed.
Response 6: We thank the Reviewer for the remark. We mention to the limitations of LSTMs for long sequences in line 268.
Comment 7: Loss functions are defined but not clearly explained in terms of practical intuition.
Response 7: Thanks for the suggestion to improve the clarity of our work. After the first introduction of the equations in the DAAE, we describe the equations intutively and relate them to the setting of Genrative Adeversarial Networks (GANs) (see line 259-262). In the equations for our joint approach, we add an interpretation for the reconstruction loss in line 295.
Comment 8: Only three lambda_adv values were tested. Lacks granularity- Why? It would be better to consider a larger quantity. In this case, the dependence on the parameter would become more apparent
Response 8: We thank the Reviewer for this remark. Thus, we added an additional experiment for a smaller lambda value than 1. The goal is to see the interplay between the Gaussian constraint and the classification performance and not to find the perfect choice. With the additional experiment, we see that choosing lambda to small or too large can lead to effects on text quality and/or accuracy.
Comment 9: I believe that the phrase "barely influences" (Line 334) is vague; how much degradation is acceptable?
Response 9: Thank you for pointing out this ambiguity. In general, the acceptable degradation is dependent on the application and we leave that to the intereseted reader.
Comment 10: The paper claims this is the first attempt at radar-text alignment no literature survey confirms that.
Response 10: Thank you for pointing us in this direction. We recently found a work called RadarLLM which was published 3 months ago, which aligns radar signal and text descriptions. However, their approach is similar to CLIP while we focus on the explainability of the classifier latent space. However, we added this work in the related work section in line 155. In addition, we rephrased this part in the abstract.
Comment 11: ROUGE is introduced without specifying which version (L, 1, 2, etc.).
Response 11: Thank for mentioning the missing specification. We added the specific ROUGE-L method in line 345.
Comment 12: The tables are formatted without regard to the journal's requirement (see template)
Response 12: We apologize for the wrong format and adjusted all the tables after reviewing the template again.
Comment 13: "We first, describe " (Line 178) contains a grammatical error comma should be removed.
Response 13: Thank you very much for mentioning this mistake. We corrected the sentence and removed the comma.
Comment 14: Grammar issue in (Line 44) - consider rephrasing.
Response 14: We thank the Reviewer for his suggestion to improve the readability. We rephrased the sentence in the revised manuscript. (see line 43-45)
Comment 15: The quality of Figure 4. should be improved.
Response 15: To improve the quality of Figure 4 we removed one of the columns and increased the resolution of the indivdual figures as well as the font size.
Comment 16: I believe that the Experiments section should be significantly expanded to demonstrate how the proposed approach performs in a more general setting. For example, when the number of people is not known in advance and varies over time. It would also be interesting to see the results obtained by the authors when the number of people is zero.
Response 16: We thank the reviewer for his insightful comment. The results for zero people are the trivial cases because the range-doppler image consists of low noise values and the text predicts "zero" most of the time. Thus, we cannot see any motion description here, which is part of the evaluation of the text. Since we show the evaluation on test recordings, the number of people is not known to the network. In addition, the test dataset conists of dedicated test recordings. Thus, the TSNE plot in Figure 5, can also be interpreted as a sequence evaluation.
In order to provide more general settings for our approach we added experiments on images from CIFAR10 and CIFAR 100 datasets in the Appendix.
Finally,
we want to express our gratitude for the reviewer's comments to improve the content and readability of our paper.
Sincerely,
Julius Ott
Reviewer 2 Report
Comments and Suggestions for Authors
The submission introduces a novel method that combines radar-based visual classification with conditional text generation by using a Denoising Adversarial Autoencoder (DAAE) to decode it into textual descriptions. The authors use FMCW radar data to classify the number of people in a room and generate corresponding natural language explanations. The paper is well-written and well-organized and as such can be considered for publication by MDPI Sensors if some minor issues are remedied.
My main concern is that the paper is completely missing the Conclusion as a formal section which should be part of every scientific paper. Please add it.
Some minor issues include:
Line 38: "The data collected by radars is..." should be "The data collected by radars are..." since in formal and technical/scientific writing "data" should be used as plural.
Line 187: "a very high distance resolution of up to 3cm..." should perhaps better be "a very high distance resolution of down to 3cm...".
Table 1: frame repetition time ts is missing.
It would be good to provide some reference for the equations (1)-(4), not just state them out of nowhere. As a suggestion, consider referencing: https://doi.org/10.2298/FUEE1804547M or similar radar literature.
Probably there is no need to break the line in equation (8).
Line 278: Better move comma after the equation (11), instead of starting the paragraph with comma.
Line 302: Define the abbreviation for Stochastic Gradient Descent (SGD).
Figure 5. Either define x1 or x2 on x-axis and y-axis or omit the axes altogether. Also, consider changing the marker shape (don't use circle) for wrong predictions with 4 and 5 people to make them better visible when printed grayscale.
Author Response
Dear Reviewer,
thank you very much for the insightful suggestions to improve our paper. In the following, we will adress each remark individually and the respective changes are highlighted in red in the revised manuscript,
Comment 1: My main concern is that the paper is completely missing the Conclusion as a formal section which should be part of every scientific paper. Please add it.
Response 1: Thank you for pointing us to this mistake. We sincerely apologize for this mistake and added the conlusion.
Some minor issues include:
Comment 2: Line 38: "The data collected by radars is..." should be "The data collected by radars are..." since in formal and technical/scientific writing "data" should be used as plural.
Response 2: Thank you very much for the suggestion. We updated the manuscript accordingly.
Comment 3: Line 187: "a very high distance resolution of up to 3cm..." should perhaps better be "a very high distance resolution of down to 3cm...".
Response 3: Thank you again for the hint. We changed the sentence in the revised manuscript.
Comment 4: Table 1: frame repetition time ts is missing.
Response 4: Thank you for pointing out the missing value. It has been added in the revised version of the paper.
Comment 5: It would be good to provide some reference for the equations (1)-(4), not just state them out of nowhere. As a suggestion, consider referencing: https://doi.org/10.2298/FUEE1804547M or similar radar literature.
Response 5: We thank the reviewer for the suggestion. We added this citation for equations (1) - (4) to support their validity.
Comment 6: Probably there is no need to break the line in equation (8).
Response 6: Thank you very much for this suggestion. We removed the line break in equation (8).
Comment 7: Line 278: Better move comma after the equation (11), instead of starting the paragraph with comma.
Response 7: In the revised manuscript, we placed the comma after equation (11). Thanks!
Comment 8: Line 302: Define the abbreviation for Stochastic Gradient Descent (SGD).
Response 8: To improve readability, we added the definition of the abbreviation for Stochastic Gradient Descent
Comment 9: Figure 5. Either define x1 or x2 on x-axis and y-axis or omit the axes altogether. Also, consider changing the marker shape (don't use circle) for wrong predictions with 4 and 5 people to make them better visible when printed grayscale.
Response 9: Thank you very much for the helpful suggestion. We changed Figure 5 such that the different types of wrong predictions are highlighted by different shapes. In addition, we omited the axis description.
Finally, we want to express our gratitude for the reviewer's comments to improve the content and readability of our paper.
Sincerely,
Julius Ott